# Influence of Temperature and De-Icing Salt on the Sedimentation of Particulate Matter in Traffic Area Runoff

**Steffen H. Rommel** and **Brigitte Helmreich** *

Chair of Urban Water Systems Engineering, Technical University of Munich, Am Coulombwall 3, 85748 Garching, Germany; s.rommel@tum.de
* Correspondence: b.helmreich@tum.de; Tel.: +49-89-289-13719

**Abstract:** Stormwater quality improvement devices use sedimentation as a pre-treatment step to separate contaminant laden particulate matter (PM) from traffic area runoff. Multiple studies describe worse settling behavior during the cold season. This paper is written in response to a decreased PM retention that was observed in the cold season during a 20-month monitoring of a sedimentation tank. However, the data was insufficient to assess the two factors that influence sedimentation during the cold season—temperature and de-icing salt application. Therefore, simplified discrete particle settling models were used to determine the influence of temperature and de-icing salt. These influences were compared to other factors, like overflow rate, particle density, and particle size distribution. To calculate the effect of temperature and de-icing salt on density and viscosity, two empirical models were applied for the first time in this field. The calculations showed that de-icing salt (NaCl) had a negligible influence on the retention of PM. However, reducing the temperature from 20 °C to 5 °C was shown to decrease the total suspended solid removal efficiency by up to 8%. The order of influencing factors was found to be particle size distribution >> overflow rate > particle density > temperature.

**Keywords:** traffic area runoff; road runoff; sedimentation; stormwater treatment; de-icing salt; cold season; viscosity; stormwater quality improvement device; SQID

## 1. Introduction

Traffic area runoff is contaminated by, amongst others, wear of pavement and vehicles, leakage, and atmospheric deposition [1–6]. Most of the contaminants, e.g., heavy metals, are present in particulate form [6].

In the winter season, abrasives and de-icing salts are applied on roads to ensure traffic safety [7,8]. This application directly causes a higher load of particulate matter (PM) to be present on the road surface [8]. This PM is associated with higher concentrations of contaminants in traffic area runoff due to higher wear and tear of pavement and vehicles, as well as an increased corrosion, enhanced by de-icing salts [9–11]. Because most contaminants are particulate bound, an effective removal of PM from traffic area runoff needs to be considered before it is discharged into groundwater or surface water.

The use of decentralized sustainable urban drainage systems (SUDS) for the treatment of stormwater runoff is becoming increasingly prevalent for the treatment of stormwater runoff at the source [12]. Sustainable urban drainage systems are particularly suitable in dense urban areas as they can provide viable alternatives to more common centralized treatment solutions, such as large end-of-line sedimentation tanks and retention-type soil filters. Among these are technical SUDS or

stormwater quality improvement devices (SQIDs), including sedimentation and filtration devices. Stormwater quality improvement devices utilize, in addition to sedimentation different treatment processes based on physical adsorption, precipitation and chemical processes [12,13]. However, their pollution removal efficiencies can be quite different [12]. In the most cases, sedimentation is the pre-treatment step of a multi-step treatment to avoid clogging of sorption filters. Nevertheless, sedimentation is sometimes the sole treatment step.

During a monitoring campaign of a SQID treating the runoff of a highly trafficked road, we observed considerably worse effluent quality of a sedimentation tank during winter [14]. Figure 1a gives an impression of road runoff (left) and effluent of the sedimentation tank (right) of a snowmelt event under de-icing salt influence. This finding is supported by Semadeni–Davies [15], who reported a decreased performance of an urban stormwater pond during winter and spring. Figure 1b highlights multiple possible influencing factors for worse effluent quality of SQIDs during the cold season adapted from Semadeni–Davies [15].

Stratification of the water column in the sedimentation tank caused by temperature differences between the influent and the water volume in the tank can lead to disadvantageous flow patterns and shorter residence times in the system [16,17]. Additionally, densimetric stratification induced by de-icing salt application is increasing this effect [18]. Tchobanoglous et al. [19] mention that as low as a 1 °C temperature differential between incoming wastewater and the water in the sedimentation tank will lead to a density current. Additionally, lower temperature and higher de-icing salt concentration increase density and viscosity of the water in the sedimentation basin. This reduces the settling velocity of particles, and therefore, lowers the removal efficiency of PM in the system. Winkler et al. [20] show that in wastewater treatment, the settling velocity of granular sludge decreases with lower temperature and higher salt (NaCl) concentrations. Similar findings can be expected for particles in urban runoff. However, the properties of granular sludge in wastewater treatment are considerably different to road runoff particles. Furthermore, studies showed that the flocculation of PM is affected by the temperature [21–23]. However due to a decreasing floc density with an increasing floc size, it is not clear if flocculation is definitely improving the settling behavior. Thus, this study is describing discrete particle settling.

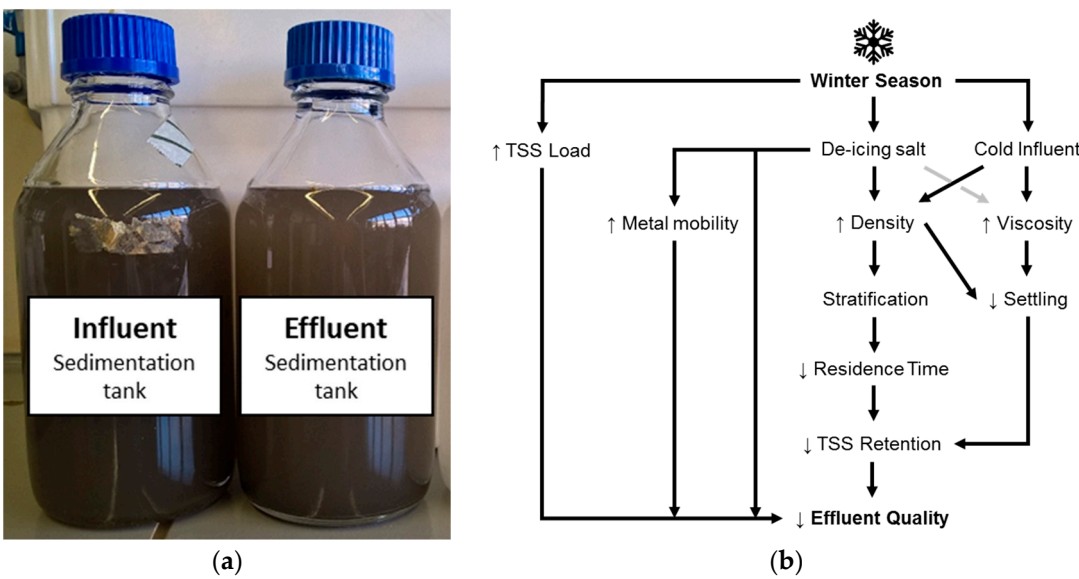

(**a**)  (**b**)

**Figure 1.** (**a**) Influent (**left**) and effluent (**right**) samples of a sedimentation tank for road runoff treatment withdrawn during the cold season. Minor difference can be seen in turbidity between the two samples. (**b**) Diagram showing which influencing factors cause worse effluent quality and total suspended solids (TSS) retention of sedimentation tanks for treatment of road runoff during winter season; grey arrow indicates weaker influence (adapted from Semadeni–Davies [15]).

Adamsson and Bergdahl [17] indicated that temperature and salinity need to be considered to evaluate the performance of detention tanks. Spelman and Sansalone [24] reported that temperature has a minor impact on the treatment efficiency of hydrodynamic separators and clarifiers. However, in this study a minimum temperature of 10 °C was used, which is not representative for the temperate climate zone. Yet, no study covered the influence of temperature and de-icing salt concentration on the settling behavior of PM in traffic area runoff. Because of the relatively low concentration of PM, measured as total suspended solids (TSS), an experimental approach would be affected by huge uncertainties. Reliable statements for a better understanding of the processes, and thus estimation of maintenance intervals is necessary to aid in increasing the acceptance of SQIDs in the future.

The aim of this study was to evaluate the ranking of influencing factors causing decreased performance during the cold season of sedimentation tanks as part of SQIDs used for treatment of traffic area runoff. After analyzing data from a 20-month monitoring of a full-scale sedimentation tank, the influencing processes based on physical settling models were evaluated to substantiate the results.

To calculate densities and viscosities of various de-icing salt solutions the models of Laliberté [25,26] were applied for the first time in this field. Future numerical models could be performed based on those models to consider temperature as well as de-icing salt effects in SQIDs.

## 2. Materials and Methods

### 2.1. Monitoring of Full-Scale Sedimentation Tank

To investigate PM removal efficiency of SQIDs for road runoff a full-scale sedimentation tank was monitored. The monitored sedimentation tank is the primary stage of a two-step SQID for road runoff located in Munich, Germany. It is operated as a continuous flow sedimentation tank. The tank consists of a round concrete shaft with a diameter of 2 m, an impounding depth of 2.4 m and a storage volume of 7.5 m$^3$. Runoff from 400 m$^2$ of a heavily trafficked road form the influent to the treatment plant. The annual average daily traffic (AADT) is approximately 24,000 vehicles per day on the attached road surface (46,000 vehicles per day including the separated opposing traffic). The cross-section of the road consists of two traffic lanes, one accelerating lane and one emergency lane with an asphalt surface.

Samples of the influent and effluent of the sedimentation tank were withdrawn time-proportionally (5 min intervals, 250 mL each) during rain events by automatic samplers (WaterSam WS 316 and Edmund Bühler PP 84). The sampling took place from May 2016 to January 2018. The sampling started after the inflow exceeded 1 L min$^{-1}$ for longer than 1 min. Flow was measured by an electro-magnetic flow meter (Krohne Optiflux 1100 C, DN40, error of measurement <1.6% for q > 2.6 L s$^{-1}$ ha$^{-1}$). At the end of the discharge events the sampling stopped. To prevent alteration of samples, they were kept in coolers at 4 ± 1 °C until transport to the lab. Analyses were performed within 72 h.

The discrete samples withdrawn during one discharge event were combined to create one composite sample. Electric conductivity (EC) and pH values were analyzed according to SM 2510 B (detection limit: 1 μS cm$^{-1}$) and SM 4500-H$^+$, respectively [27]. Electric conductivity was used to verify the de-icing salt application, here sodium chloride (NaCl). Particulate matter was analyzed as total suspended solids (TSS), coarse suspended solids with a diameter greater than 63 μm (SS > 63 μm), and suspended solids between 0.45 μm and 63 μm (SS63). To determine the PM, one liter of sample was sieved by a 1000 μm sieve in the first step, followed by a 63 μm sieve as the second step. In the third step the sieved sample was filtrated under a vacuum over a 0.45 μm membrane filter (cellulose nitrate). Large constituents (>> 1 mm; e.g., leaves, cigarette stubs), which are not representative for the sample, were manually removed. All sieves and filters were dried at 105 °C ± 2 °C until constant mass was achieved. The fine suspended solids (SS63) are the fraction between 0.45 μm and 63 μm, found as residue on the membrane filter. The coarse suspended solids (SS > 63) are the fraction between 1000 μm and 63 μm, found on the 63 μm sieve. Total suspended solids was calculated by the sum of the residues

on the 1000 µm and 63 µm sieves, and the 0.45 µm membrane filter after drying. The procedure was a modified method of Dierschke and Welker [28].

Climate data of the measurement station 3379 of The German Weather Service (DWD) was used. The meteorological station is located approximately 5 km from the study site. The annual precipitation height was 955 mm in 2016 and 931 mm in 2017. In 2016, there were 64 days of frost (minimum air temperature < 0 °C) and 71 days in 2017. The temperature of the water in the sedimentation tank was assumed to be equal to the soil temperature at 50 cm depth.

### 2.2. Calculations

#### 2.2.1. Density

Densities of water $\rho_w$ (kg m$^{-3}$) were calculated with Equation (1) [25],

$$\rho_w = \frac{\left(\left(\left(\left(\left(-\,2.8054253 \times 10^{-10}t + 1.0556302 \times 10^{-7}\right)t - 4.6170461 \times 10^{-5}\right)t - 0.0079870401\right)t + 16.945176\right)t + 999.83952\right)}{1 + 0.01687985t} \tag{1}$$

where t is the temperature (°C). The densities of solutions $\rho_m$, in this case sodium chloride (NaCl) solutions, were calculated using the model of Laliberté and Cooper [25]. The model considers temperature and concentration of the solute. $\rho_m$ (kg m$^{-3}$) was determined with the following Equations (2) and (3):

$$\rho_m = \frac{1}{\frac{w_{H_2O}}{\rho_{H_2O}} + w_{NaCl}\,\overline{v}_{app,NaCl}} \tag{2}$$

$$\overline{v}_{app,NaCl} = \frac{w_{NaCl} + 1.01660 + 0.014624\,t}{(-0.00433\,w_{NaCl} + 0.06471)e^{(0.000001(t+3315.6)^2)}} \tag{3}$$

where $w_{H_2O}$ = mass fraction of water in the solution (-), $w_{NaCl}$ = mass fraction of the NaCl in the solution (-) and $\overline{v}_{app,NaCl}$ = NaCl specific volume (m$^3$ kg$^{-1}$). Equation (3) is valid for $w_{NaCl} > 0.00006$ [25], therefore densities of pure water (without NaCl) were calculated with Equation (1).

This study examines the de-icing salt NaCl concerning the by far greatest application rate [7,29]. Concentrations of up to 15 g L$^{-1}$ NaCl could be present in traffic area runoff [7,29,30]. Because of the thermal expansivity, this study used the mass fraction of solute NaCl ($w_{NaCl}$) to describe the concentration of the solution. A $w_{NaCl}$ of 0.01 corresponds to 10 g L$^{-1}$ at 20 °C. To cover extrema, the calculations were conducted up to $w_{NaCl} = 0.02$, unless otherwise stated.

#### 2.2.2. Viscosity

Dynamic viscosities of the solutions were calculated as a function of temperature and concentration of the solute by means of the model of Laliberté [26], following Equation (4). In the first step the dynamic viscosity of water $\eta_w$ (kg m$^{-1}$ s$^{-1}$) was calculated at temperature t (°C). Equation (4) is valid for t from 0 °C to 100 °C [26].

$$\eta_w = \frac{(t + 246) \times 10^{-3}}{(0.05594t + 5.2842)t + 137.37} \tag{4}$$

In the second step the viscosity of the various NaCl solutions $\eta_m$ (kg m$^{-1}$ s$^{-1}$) were calculated with Equation (5), based on the model of Laliberté [26]:

$$\eta_m = \eta_w^{w_{H_2O}}\left(\exp\left(\frac{16.222(1 - w_{H_2O})^{1.3229} + 1.4849}{(0.0074691t + 1)\left(30.78(1 - w_{H_2O})^{2.0583} + 1\right)}\right) \times 10^{-3}\right)^{w_{NaCl}}. \tag{5}$$

where $\eta_m$ = dynamic viscosity of the NaCl solution (kg m$^{-1}$ s$^{-1}$), $\eta_w$ = dynamic viscosity of water (kg m$^{-1}$ s$^{-1}$), $w_{H_2O}$ = mass fraction of water in the solution (−), $w_{NaCl}$ = mass fraction of NaCl in the solution (−) and t = temperature (°C). This formula is valid for t ranging from 5 °C to 154 °C and $w_{NaCl}$ up to 0.264 [26].

### 2.2.3. Settling Velocity

Stokes' Law was used to determine the terminal settling velocity $v_t$ (m s$^{-1}$) of the particles [31],

$$v_t = \frac{g(\rho_s - \rho_w)d^2}{18\eta_m} \tag{6}$$

where g = gravitational acceleration (9.81 m s$^{-2}$), $\eta_m$ = dynamic viscosity of the solution (kg m$^{-1}$ s$^{-1}$), $\rho_m$ = density of the solution (kg m$^{-3}$), $\rho_s$ = density of particles (kg m$^{-3}$), and d = particle diameter (m).

Because Equation (6) is only applicable for laminar flow (Reynold's number Re < 1) [19], Re was calculated afterwards with Equation (7),

$$Re = \frac{v_t d}{\nu_m} \tag{7}$$

where $\nu_m$ = kinematic viscosity of the solution (m$^2$ s$^{-1}$). Reported settling velocities fulfilled this condition, meaning laminar flow conditions were present.

The mostly non-spherical shape of road runoff particles [32,33] was neglected due to a lack of data about sphericity factors. To investigate the influence of particle density, a low $\rho_s$ (1.35 g cm$^{-3}$) [34] and a high $\rho_s$ (2.25 g cm$^{-3}$) [33] were used. Because fine SS63 are prevalent in traffic area runoff [30,35–37] and showed a significant worse settling behavior, this study focused on those particles <63 µm.

### 2.2.4. Retention of Suspended Solids

To determine the retention of suspended solids a stationary idealized discrete settling approach was selected. Inflow $Q_{In}$ and outflow $Q_{Eff}$ were assumed to be equal and constant over time as well as inflow TSS. Scouring of retained particles was neglected. In this model particles with a terminal settling velocity equal to or greater than the critical terminal settling velocity $v_{crit}$ were removed [19]. The $v_{crit}$ (m s$^{-1}$) was equal to the overflow rate determined with Equation (8),

$$v_{crit} = \frac{Q}{A} = overflow\ rate \tag{8}$$

where Q (m$^3$ s$^{-1}$) = flow rate and A (m$^2$) = surface of the sedimentation basin.

By means of $v_{crit}$ the critical particle diameter $d_{crit}$ (µm) was calculated for given conditions like solution temperature, viscosity, density, and particle density. The calculation was conducted with Excel Solver and the solving method GRG nonlinear. Because de-icing salt (NaCl) showed a neglectable effect on the settling velocity (cf. Section 3.2.3), in this step $w_{NaCl}$ was not considered (set to 0). The water temperature was altered in the range from 5 °C to 25 °C. Additionally, the effect of particle density was studied.

The model system corresponded to the monitored system (cf. Section 2.1). The discharge coefficient $\psi$ was set to 1.0, to represent a fully impervious surface and ensure comparability to the test protocol of the Deutsches Insitut für Bautechnik (DIBt) [38]. Rain intensities of 2.5 L s$^{-1}$ ha$^{-1}$ and 15 L s$^{-1}$ ha$^{-1}$ were investigated. The upper value was chosen based on the knowledge that more intense design rain intensities lead to a minor increase in treatment efficiency [39].

In real traffic area runoff, particles are not uniform in particle size, and therefore, the particle size distribution (PSD) was characterized by a parametric cumulative distribution function (CDF). Selbig and Fienen [40] proposed the usage of the Rosin–Rammler distribution, which is equivalent to

the Weibull distribution. This approach was continued by Lee et al. [41]. Based on this knowledge, this method was used to generate a synthetic PSD. The formula of the Rosin–Rammler CDF is

$$W(d, \lambda, \kappa) \ = \ 1 - e^{\left(-\frac{d}{\lambda}\right)^{\kappa}} \tag{9}$$

where $W(d, \lambda, \kappa)$ = mass fraction less than d (−); d = particle diameter (μm); $\lambda$ = shape parameter (−); $\kappa$ = scale parameter (−) [40]. For our simulation we used $\lambda$ = {9.3, 60, 116}, equal to a $d_{50}$ from 6 to 75 μm, and $\kappa$ was fixed to 0.84 [41].

The TSS removal efficiency (−) was calculated with Equation (10) based on the critical particle diameter $d_{crit}$, which is derived from $v_{crit}$ following Equation (8). Preassigned $\lambda$ and $\kappa$ values were used.

$$TSS \ removal \ efficiency \ = \ 1 - W(d_{crit}, \lambda, \kappa) \tag{10}$$

## 3. Results and Discussion

### 3.1. Monitoring of the Full-Scale Sedimentation Tank

The monitoring of the full-scale sedimentation tank was executed over a period of 20 months. To cover all seasonal influences, 23 rain events were taken and analyzed. Table 1 presents the statistics of all measured values from the influent and effluent of the sedimentation tank as well as the calculated retention of TSS, SS > 63, and SS63. Due to the flushing of PM of former events, a few negative removal rates were determined. Remarkable was that the PM occurred almost completely as SS63 (98% of mean TSS). Thereby the focus on the fine fraction of this study was supported.

**Table 1.** Statistics of full-scale sedimentation tank monitoring: q = mean inflow; t = water temperature in sedimentation tank; TSS Inf. = influent TSS; SS63 Inf. = influent SS63; EC Eff. = electric conductivity in effluent.

| Parameter | Unit | *n* | Min | 25th Percentile | Median | Mean | 75th Percentile | Max | SD |
|---|---|---|---|---|---|---|---|---|---|
| q | L s$^{-1}$ ha$^{-1}$ | 23 | 1.6 | 3.4 | 5.7 | 8.1 | 7.2 | 26.1 | 7.5 |
| t | °C | 23 | 3.0 | 8.8 | 17.5 | 17.5 | 21.7 | 24.6 | 7.3 |
| TSS Inf. | mg L$^{-1}$ | 23 | 7 | 32 | 82 | 123 | 166 | 433 | 129 |
| SS > 63 Inf. | mg L$^{-1}$ | 15 | 3 | 18 | 26 | 35 | 46 | 122 | 30 |
| SS63 Inf. | mg L$^{-1}$ | 15 | 8 | 18 | 73.3 | 121 | 140 | 394 | 130 |
| EC Eff. | μS cm$^{-1}$ | 22 | 59 | 95 | 119 | 119 | 867 | 4790 | 1225 |
| TSS retention | % | 21 | −365 | 22 | 44 | 18 | 68 | 95 | 105 |
| SS > 63 retention | % | 15 | −4 | 65 | 83 | 71 | 93 | 100 | 31 |
| SS63 retention | % | 15 | −49 | −1 | 37 | 25 | 57 | 80 | 40 |

Figure 2 shows the annual course of TSS influent and effluent of the sedimentation tank, as well as TSS retention. Due to strong seasonal courses [10], data points of different years are plotted together in one figure. At the start of the sampling campaign two events showed strong scouring of sediments (−185% and −365% TSS retention) from previous rain events or maybe even residues of the construction work. Therefore, those two events were removed. In the annual course of TSS the strong increase in the cold seasons was recognizable. Likewise, a decrease of TSS retention during the cold season is shown. Furthermore, in summer three events showed conspicuous low TSS retentions (1%, 2%, and 28%). The event with the lowest TSS retention was the event with the highest inflow. Thus, the high inflow resulted in the low TSS retention. Although the data was not adequate to explain the reason for the low TSS retentions of the other two events. Due to less frequent and less intensive rain events in the cold season, only a few events could be sampled. Therefore, monitoring will be continued to gain more statistical significance.

To evaluate the influence of temperature on the PSD, Figure 3 shows the ratio SS63/TSS in the influent and effluent of the sedimentation tank. During the cold season, the particles were smaller than during summer. Thus, a decreased settling velocity was expected during the cold season. Because

SS63/TSS was higher in the effluent than in the influent, the worse removal efficiency of SS63 was visible. The increasing particle size with increasing temperature can be explained by enhanced flocculation [21,22]. Since SS63 is describing a range of particle sizes, the ration SS63/TSS can only be an indicator for the PSD.

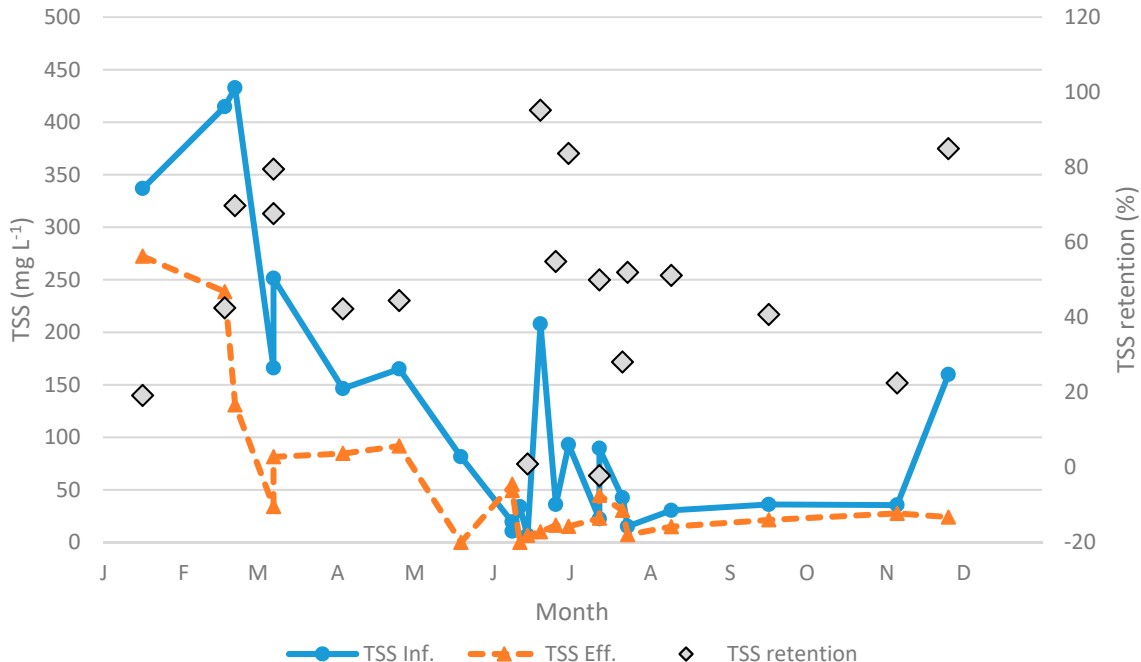

**Figure 2.** Annual course of TSS influent and effluent of the sedimentation tank and TSS retention; two successive data points of TSS retention were removed due to strong scouring of sediments of previous events (−185% and −365% TSS retention).

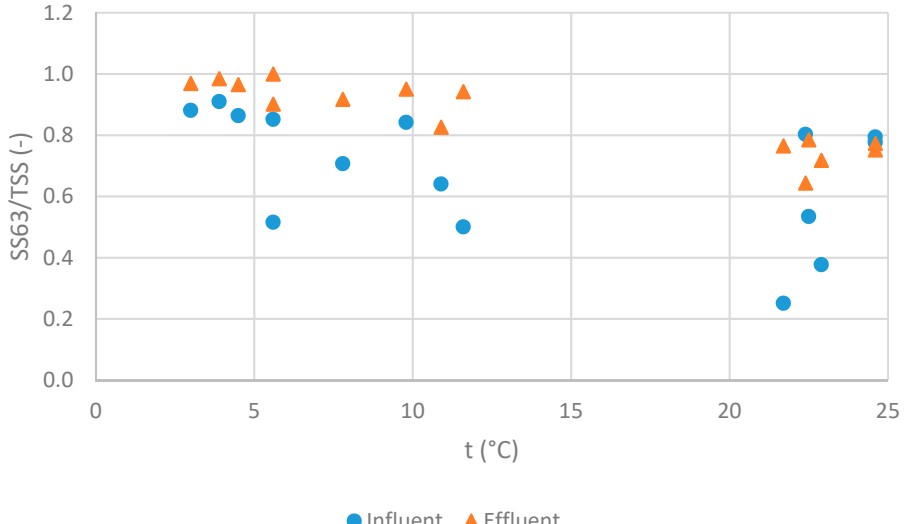

**Figure 3.** Fraction of fine particulate matter (PM) (SS63) in the TSS in the influent and effluent of the sedimentation tank as a function of temperature.

The following correlation matrix (Figure 4) shows multiple effects influencing the settling of PM. During cold temperatures, a higher EC, due to the application of de-icing salt, and influent TSS was present ($r_s < -0.68$). Between SS63 removal and mean inflow q a weak correlation was observable($r_s = -0.34$), which induces that even longer residence times in the studied sedimentation tank do not improve the SS63 retention. In contrast, the retention of SS > 63 correlates considerably with q ($r_s = -0.71$). Overall, this means that the used settling tank is only partially suitable for the

retention of fine particulate matter. Only in recent years the SS63 parameter has become popular, and therefore settling tanks were mainly designed to separate the sand fraction (>63 μm). To achieve better settling, the geometry needs to be changed to achieve lower surface loadings, and thus, increased settling times.

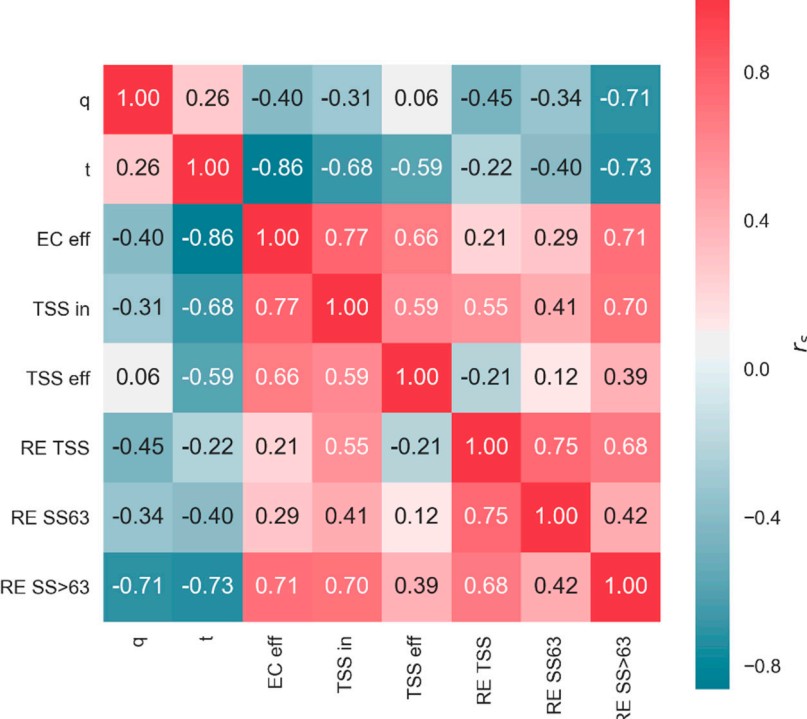

**Figure 4.** Correlation matrix of Spearman's rank correlation coefficients $r_s$ of full-scale sedimentation tank monitoring; q = mean inflow (L s$^{-1}$ ha$^{-1}$), t = water temperature in tank (°C), EC eff. = effluent electrical conductivity (μS cm$^{-1}$), TSS in = influent TSS (mg L$^{-1}$), TSS eff = effluent TSS (mg L$^{-1}$), RE TSS = removal efficiency of TSS (%), RE SS63 = removal efficiency of SS63 (%), RE SS63 = removal efficiency of SS > 63 (%), *n* = 15 for SS63 and *n* = 23 for all other parameters.

Although it was expected that TSS, SS63, and SS > 63 retentions would increase with increasing temperature, following sedimentation theory (cf. Section 2.2), this was not observable in the data. However, an inverse effect is shown in Figure 4, which conflicts with the physical basics of settling. Due to the complex nature of the system, it looks like there is a spurious correlation between temperature and the PM retention. The same unexpected correlation is visible between EC and PM retention. Though TSS and SS63 retention correlate moderately to weakly with t and EC ($|r_s| \leq 0.40$). Because q is increasing with t and decreasing with EC, it can be assumed that this is masking the expected effects. In addition, PSD was not constant (cf. Figure 3), and therefore removal efficiency of the PM was varying. The monitoring will proceed and with increasing data the trend may alter.

Given that our findings are based on a limited number of sampled events and the uncertainty of sampling, the results of that monitoring should be treated with caution. Bardin et al. [42] demonstrated with a comparable system that established methods of performance measurement are accompanied with a wide margin of uncertainty. Assuming an ideal sampling and preservation of the samples, the removal efficiency of TSS is already within a range of 39% to 59% due to the error of TSS analysis (±10%). This approximation is based on the mean TSS concentrations in the influent and the effluent of the sedimentation tank. Furthermore, there are uncertainties related to the sampling conditions, the sampling cycle performance and the preparation of the composite samples [42].

To further evaluate the ranking of influencing factors on PM retention, the following calculations, based on simplified approaches, were made to substantiate the results.

*3.2. Calculations*

3.2.1. Density

The densities of various NaCl solutions were calculated with the model of Laliberté and Cooper [25]. Figure 5a shows the calculated densities of solutions $\rho_m$ with different $w_{NaCl}$ at various temperatures. The solution with $w_{NaCl} = 0.00$ equates to pure water. Density declines with increasing temperature and increases with increasing $w_{NaCl}$. The density anomaly of water affected the NaCl solutions as well. While the highest density of pure water was found at 3.98 °C, in accordance with Kell [43], the density of the NaCl solutions peaked at lower temperatures: 1.58 °C with $w_{NaCl} = 0.02$ and up to 3.38 °C with $w_{NaCl} = 0.005$. Thereby a trend of shifting the maximum density point towards lower temperatures with increasing $w_{NaCl}$ was derivable. In Figure 5b, density is plotted as a function of $w_{NaCl}$. A linear correlation between $\rho_m$ and $w_{NaCl}$ is observable. Density affects settling velocity linearly following Equation (6) (Section 2.2.2), causing colder temperatures and higher $w_{NaCl}$ to lead to a decrease of settling velocity and consequently worse PM retention. Assuming the particle density of $\rho_s = 1.35$ g cm$^{-1}$, the settling velocity at 5 °C and $w_{NaCl} = 0.02$ is 5% less than at 20 °C and $w_{NaCl} = 0.00$. Though, $w_{NaCl}$ is an extreme value in this example. Under more likely winter conditions ($w_{NaCl} = 0.01$), the effect of density variation on settling velocity is <1%. Consequently, this effect is negligible and probably not measurable in full scale. Please refer the Supplementary Materials (S1) for the densities of NaCl solutions at various temperatures.

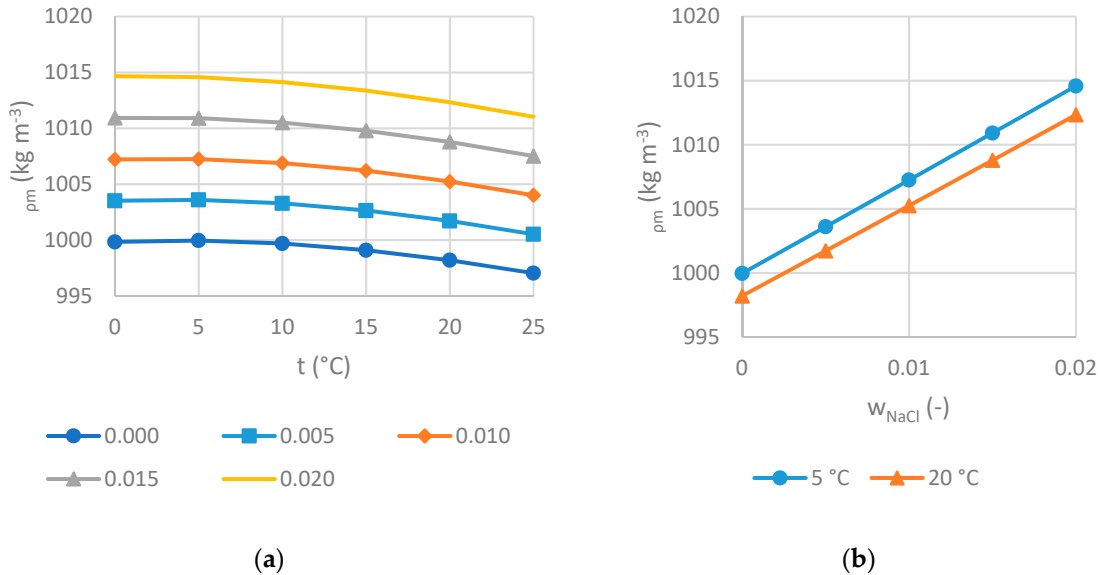

(**a**)                                                   (**b**)

**Figure 5.** (**a**) Density of aqueous NaCl solutions as a function of temperature for various $w_{NaCl}$; (**b**) density of aqueous NaCl solutions as a function of $w_{NaCl}$ for t = 5 °C and t = 20 °C.

3.2.2. Viscosity

Based on the model of Laliberté [26], the viscosity of various NaCl solutions were calculated at 20 °C. Figure 6a shows the viscosity as a function of the mass fraction of NaCl in the solution $w_{NaCl}$. NaCl showed a kosmotropic effect, meaning it acts as a structure maker and increases the viscosity of the solution compared to pure water [44]. The viscosity of pure water is depicted by the solution with $w_{NaCl} = 0.00$.

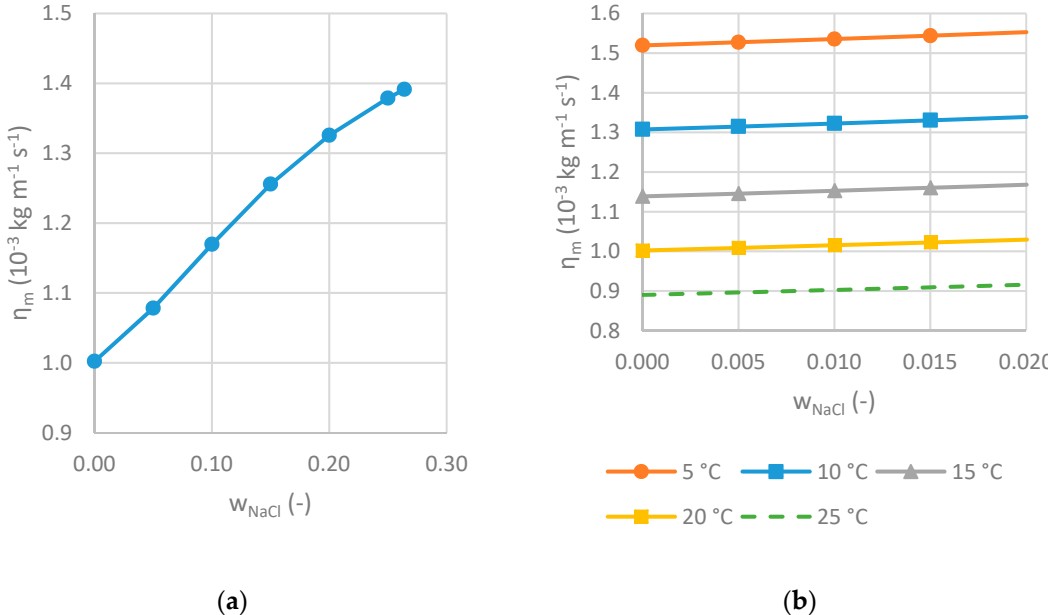

**Figure 6.** (**a**) Viscosity of various aqueous NaCl solutions at 20 °C; (**b**) viscosity of aqueous NaCl solutions at various temperatures.

In traffic area runoff values of $w_{NaCl} < 0.015$ (15 g L$^{-1}$ at 5 to 25 °C) occur [7,30]. Consequently, the viscosity alteration by NaCl does not occur in the full magnitude, which Figure 6a indicates. To investigate the influence of the temperature, the viscosity calculation was carried out with solution temperatures t from 5 to 25 °C with increments of 5 °C following the method of Laliberté [26]. The minimal temperature covered by the model is 5 °C, thus the density peak (cf. Section 3.2.1) cannot be included in the viscosity calculation. Figure 6b shows the viscosity of solutions $\eta_m$ as a function of $w_{NaCl}$ at various temperatures. In the considered mass fraction range, a slightly positive linear relation between $w_{NaCl}$ and $\eta_m$ is observable. However, the influence of the solution temperature was significantly stronger. Thereby the solutions revealed a higher viscosity at lower temperatures. The settling velocity in a solution with $w_{NaCl} = 0.02$ at 5 °C is 35% less than in a solution with $w_{NaCl} = 0.00$ at 20 °C. Accordingly, viscosity is the main influencing factor on the settling velocity if particle density and size are constant. The viscosity variation was mainly caused by temperature alteration, not by de-icing salt (NaCl). Please refer the Supplementary Materials (S1) for the viscosities of NaCl solutions at various temperatures.

### 3.2.3. Settling Velocity

To assess the influence of temperature and de-icing salt on the settling velocity two scenarios were assumed: Winter conditions with t = 5 °C and $w_{NaCl} = 0.02$ and summer conditions with t = 20 °C and $w_{NaCl} = 0.00$. Figure 7 illustrates settling velocity $v_t$ as a function of particle diameter d. There was a severe difference observable between both scenarios. In winter, 38% lower settling velocities were determined.

To analyze the influence of de-icing salt (NaCl) another scenario was modeled at 5 °C solution temperature with $w_{NaCl} = 0.00$. In this case, a 34% lower settling velocity was calculated for winter conditions without NaCl influence. This resulted in the temperature influencing the settling velocity at a bigger magnitude than the de-icing salt.

Based on that knowledge, it is recommended to design sedimentation tanks with a larger surface area, and thus, lower overflow rates to cope with challenging winter conditions.

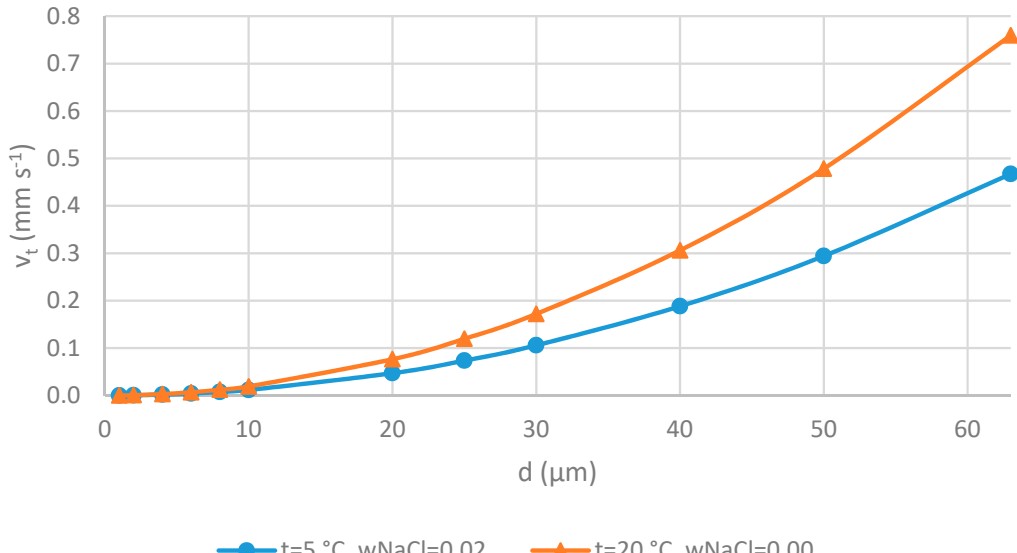

**Figure 7.** Settling velocity as a function of particle diameter in two aqueous solutions at various temperatures and $w_{NaCl}$ simulating winter and summer conditions (extrema); both $\rho_s$ = 1.35 g cm$^{-1}$.

### 3.2.4. Retention of Suspended Solids

Based on the monitored sedimentation tank (cf. Section 2.1) critical particle diameters $d_{crit}$ under varying boundary conditions (temperature, flow q, particle density) were determined. The critical particle diameter characterizes the lower limit of particle size, which can be separated in the sedimentation tank.

Figure 8 shows that the influence of the temperature on the particle retention is more severe with increasing flow. Furthermore, the influence of the particle density on the separation increases with flow. To assess the TSS removal efficiency PSD was considered. Hereby the information of Figure 8 was relativized. A reduced temperature from 25 to 5 °C lead to differences of ≤8% TSS removal efficiency (Figure 9a).

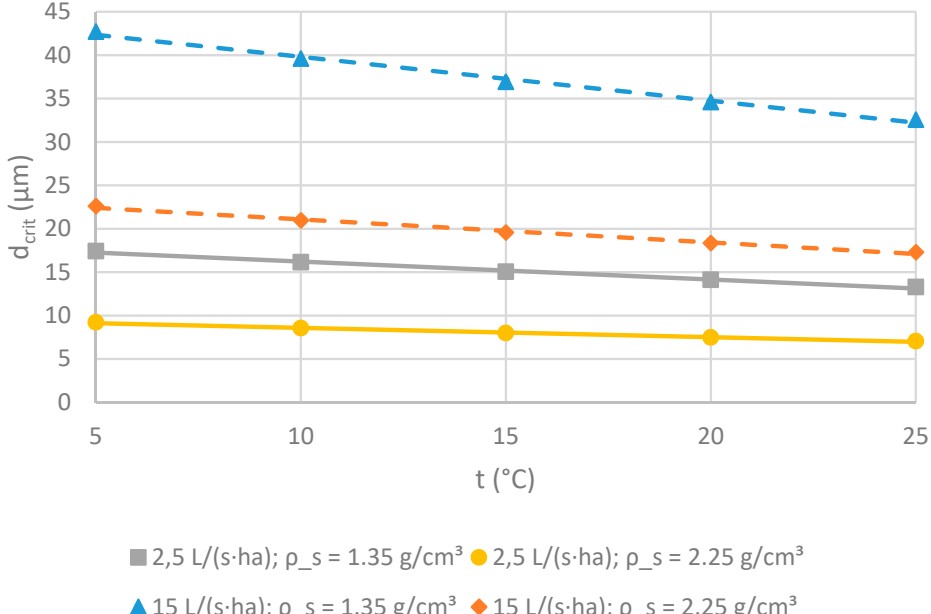

**Figure 8.** Critical particle diameter $d_{crit}$ for settling, various water temperatures and particle densities.

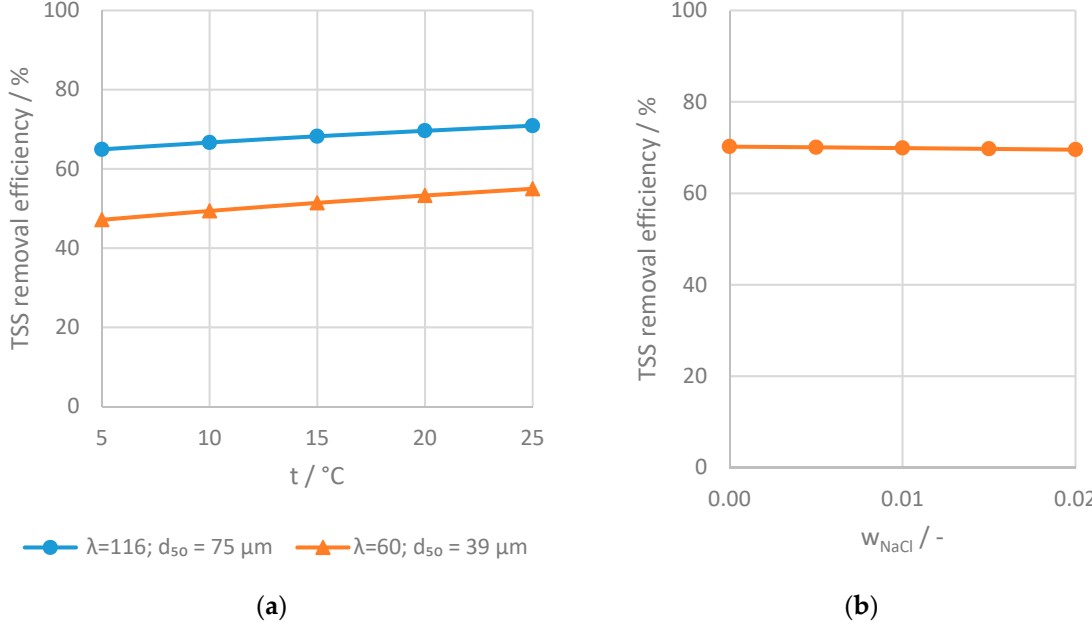

**Figure 9.** (**a**) TSS removal efficiency as a function of temperature; (**b**) TSS removal efficiency as a function of $w_{NaCl}$, $\lambda = 60$, $d_{50} = 39$ μm; (**a**,**b**): $\rho_s = 1.35$ g cm$^{-1}$; $q = 15$ L s$^{-1}$ ha$^{-1}$.

A decisive influence of de-icing salt on the TSS removal efficiency was not observable (Figure 9b). Increasing $w_{NaCl}$ from 0.00 to 0.02 was shown to decrease the TSS removal efficiency by less than 0.7%. Therefore, de-icing salt was classified as a negligible factor, which was not considered in the following ranking of the influencing factors.

The Pearson correlation coefficients were calculated based on various scenarios (t, $\rho_s$, $\lambda$ or $d_{50}$ and q) (Table 2). This allowed the influencing factors on the TSS removal efficiency to be ranked. The PSD, specified by $d_{50}$ or $\lambda$, showed the most distinct influence, followed by q and $\rho_s$. t was classified as the least influencing factor. However, physical characteristics of particles ($\rho_s$ and $d_{50}$/PSD) are regarded site-specific [45–49]. Thus, temperature needs to be considered in future SQID designs to improve treatment efficiency during the cold season. Because the applied model cannot represent effects of occurring density currents caused by temperature and de-icing salt, the effect could occur at bigger magnitudes.

**Table 2.** Pearson correlation coefficients between TSS removal efficiency, temperature t, particle density $\rho_s$, median particle diameter $d_{50}$ and discharge rate q.

|  | t | $\rho_s$ | $d_{50}$ | q |
|---|---|---|---|---|
| **TSS removal efficiency** | 0.07 | 0.22 | 0.85 | −0.32 |

The ranking of the influencing factors PSD > $\rho_s$ > t of this study was affirmed by Spelman and Sansalone [24]. Additionally, they identified influent hydrograph unsteadiness as the most powerful influencing factor. The hydrograph unsteadiness describes the shape of the hydrograph. A highly unsteady hydrograph rises fast and drops fast after a short time. In comparison to this, a highly steady hydrograph rises slow and drops slow after a long time. Due to the stationary method used in this study, this factor was not assessed. Spelman and Sansalone [24] altered the flow rate q together with the unsteadiness of the influent hydrograph, and therefore, it is not determinable if q or the unsteadiness was the dominant factor. Furthermore, the minimum temperature was 10 °C [24], which is not adequate for the use in the temperate climate zone.

To approximate realistic conditions coming from lab-scale experiments, a variety of effects were neglected in this present study. Kayhanian et al. [33] showed that particles are neither smooth nor

spherical, and therefore, models need to be established considering this influence. The current knowledge is that particles in road runoff are not spherical; however, there is no sufficient data describing the shape of runoff particles. Therefore, a representation in simulations is currently not possible. Furthermore, particles were assumed to be non-cohesive. Kayhanian et al. [33] derived from zeta-potential analysis that runoff particles have a relatively low tendency to aggregate. However, experience of handling runoff samples in the lab showed us that aggregation of particles does occur. Li et al. [50] proved this assumption with PSD analysis. Furthermore, the influence of temperature on the flocculation of PM [21–23] is not considered, yet. In addition, the influence of de-icing salt on the flocculation of runoff particles is still open. Faltermaier et al. [51] showed an increased settling velocity under de-icing influence. Studies about settling of marine clay-size sediments reported a positive correlation between salinity and settling velocity [52,53]. Since the mineralogy of the particles does have an influence on flocculation, those findings may have limited adaptability to road runoff particles, which contain clay minerals in minor quantities [46,54]. Slight variation in particle density can affect the removal efficiency of PM in sedimentation tanks. Therefore, the common method of particle density determination needs to be verified. Wet particle density [34] could reflect more realistic conditions. It can be expected that the above-mentioned uncertainties affect smaller particles more intensely than bigger size fractions [35,55]. In addition, future studies should be conducted under non-stationary conditions to consider the unsteadiness of the influent hydrograph, like that proposed by Spelman and Sansalone [24].

Further works should improve knowledge about the aforementioned aspects to improve particle separation, and therefore, achieve better effluent quality of SQIDs.

## 4. Conclusions

A sedimentation tank treating road runoff was monitored for 20 months. During the cold season, reduced PM removal efficiency was observed. However, the data was not sufficient to distinctively assess the influence of temperature and de-icing salt. Therefore, simplified settling models were applied to determine which of the influencing factors had the greatest effect on PM removal. The determined order was PSD $\gg$ q > $\rho_s$ > t. The influence of de-icing salt (NaCl) on the sedimentation of PM was negligible. Since PSD and $\rho_s$ are assumed to be site-specific, low temperatures need to be considered to improve effluent quality of SQIDs in the cold season. Low temperatures (5 $^\circ$C) revealed a decrease of up to 8% TSS removal efficiency compared to higher temperatures (20 $^\circ$C). The simplified models can be extended in future studies by considering de-icing salt induced particle coagulation, stratification, alternated flow patterns, and non-spherical shape of particles.

Two empirical models were applied the first time in this field to calculate density and viscosity of various solutions as a function of temperature and the solute concentration of de-icing salt. These seem promising and can improve future numerical models by considering non-steady water matrices.

Based on the knowledge gained about the sedimentation of PM from road runoff under cold season conditions, we recommend considering low temperatures when designing the sedimentation stages of SQIDs. Effluent quality can be thereby improved. By minimizing the PM load in the effluent of the sedimentation stage, clogging of optional downstream filtration elements can be retarded. Consequently, the intervals between maintenance events could be prolonged.

**Supplementary Materials:** The following are available online at http://www.mdpi.com/2073-4441/10/12/1738/s1, Table S1: Density and viscosity of NaCl solutions at various temperatures.

**Author Contributions:** Conceptualization, S.H.R. and B.H.; Methodology, S.H.R., Investigation, S.H.R. and B.H.; Resources, B.H.; Data Curation, S.H.R.; Writing—Original Draft Preparation, S.H.R.; Writing—Review & Editing, B.H.; Visualization, S.H.R.; Supervision, B.H.; Project Administration, B.H. and S.H.R.; Funding Acquisition, B.H.

**Funding:** This research was funded by the City of Munich.

**Acknowledgments:** This work was supported by the German Research Foundation (DFG) and the Technical University of Munich (TUM) in the framework of the Open Access Publishing Program. We thank Philipp Stinshoff of the Chair of Urban Water Systems Engineering, Technical University of Munich, for the assistance in operating the study site. We especially thank Claire Sembera for editing the draft.

**Conflicts of Interest:** The authors declare no conflict of interest.

## Abbreviations

The following symbols are used in this paper:

| | |
|---|---|
| A | Area |
| AADT | Annual average daily traffic |
| CDF | Cumulative distribution function |
| d | Particle diameter |
| $d_{50}$ | Mass-median-diameter of particles |
| DIBt | Deutsches Insitut für Bautechnik |
| DWD | German Weather Service |
| EC | Electric conductivity |
| Eff | Effluent |
| g | Gravitational acceleration |
| Inf | Influent |
| NaCl | Sodium chloride |
| PM | Particulate matter |
| PSD | Particle size distribution |
| Q | Flow rate |
| q | Discharge rate, mean inflow |
| $Q_{Eff}$ | Outflow |
| $Q_{In}$ | Inflow |
| RE | Removal efficiency |
| Re | Reynold's number |
| $r_s$ | Spearman's rank correlation coefficients |
| SQID | Stormwater quality improvement device |
| SS > 63 | Suspended solids with particle diameter >63 μm |
| SS63 | Suspended solids with particle diameter between 0.45 μm and 63 μm |
| SUDS | Sustainable urban drainage systems |
| t | Temperature |
| TSS | Total suspended solids |
| $\overline{v}_{app,NaCl}$ | NaCl specific volume |
| $v_t$ | Terminal settling velocity |
| $w_{H2O}$ | Mass fraction of water in the solution |
| $w_{NaCl}$ | Mass fraction of the NaCl in the solution |
| $\eta_m$ | Dynamic viscosity of the NaCl solution |
| $\kappa$ | Scale parameter of CDF |
| $\lambda$ | Shape parameter of CDF |
| $\nu_m$ | Kinematic viscosity of the solution |
| $\rho_s$ | Particle density |
| $\rho_w$ | Water density |
| $\psi$ | Discharge coefficient |

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
