# Peer review of "Influence of Temperature and De-Icing Salt on the Sedimentation of Particulate Matter in Traffic Area Runoff"

_water, doi:10.3390/w10121738_

Round 1

Reviewer 1 Report

General review The authors offered a timely contribution on settling of stormwater solids under various temperature and salt presence conditions. The issue is addressed rigorously, with well- demonstrated knowledge of the literature on this topic and the use of appropriate modern methods. They produced new knowledge in the field and reached the conclusions, which are well supported by the presented results. Under these circumstances, the reviewer recommends the acceptance of the paper, pending some relatively minor comments, which would be best addressed in the form of discussion. Addressing these comments should improve the paper with respect to clarity and the impact on readers. Minor comments 1. Overall the writing style is somewhat “choppy”, but clear and there is no need to make any changes in this regard. 2. P. 2, 3rd paragraph, line 4 – accumulation of chloride in settling tanks (as in stormwater ponds) would cause stronger densimetric effects than similar. 3. Fig. 1b – one influential factor missing is particle flocculation (studied e.g. by Krishnappan, B.G. and Marsalek, J. (2002). Modelling of flocculation and transport of cohesive sediment from an on-stream stormwater detention pond. Wat. Res., 36(15): 3849-3859; or Krishnappan, B.G., Marsalek, J., Watt, W.E., Anderson, B.C. (1999). Seasonal size distribution of suspended solids in a stormwater management pond. Wat. Sci. tech., 39(2), 127-134). In other words, the authors should state up front that they dealing with “discrete” particle settling. Flocculation would be typical for warmer seasons and contributes to greater sizes of particles in eq.(6). 4. P. 3, 2.1, 2nd paragraph – Flow measurement – what was the accuracy of this instrument? Note that the threshold inflow was extremely low - just 1.67 x 10-5 (L/s). 5. “To prevent alteration of samples, they were kept in coolers at 4+/-1 C… 6. The same section, paragraph 4: “ were combined to create one composite sample…” – this is a better word. 7. The same paragraph – was the upper limit of particle sizes in TSS samples applied? 8. P. 5, eq. (8) – this is also known as the equation describing the “ideal settling basin” introduced by Camp. 9. P. 6, the paragraph below Table 1, please clarify the reasoning why “Furthermore, in summer, some events showed a low TSS retention.” 10. Fig. 2, could high retentions of TSS be influenced by particle flocculation? 11. The first paragraph under Fig. 2, please reword “However, the SS63 parameter has become popular only…” 12. Paragraph 1 below table 2, the reader would benefit from seeing some explanation of “the influent hydrograph unsteadiness”. 13. Notation of Figs. 4-8, units indicated by fractions (e.g., d /µm) – somewhat unusual. Are they acceptable in this journal? I am sued to seeing units in the brackets, e.g., d(µm). 14. TSS retention efficiency – this parameter would deserve a bit of discussion of uncertainties in the present values. Such concerns were expressed e.g. by Bardin, J.P., Barraud, S. Chocat, B. (2001). Uncertainty in measuring the event pollutant removal performance of online detention tanks with permanent outflow. Urban Water, 3(1-2), 91-106. 15. P. 17, 2nd line, “sedimentation tank” was earlier called a shaft. 16. German references – it is my experience that the authors need to identify the language, and an English translation should follow the title in the original language.

Author Response

Reviewer 1

Comments:

General review The authors offered a timely contribution on settling of stormwater solids under various temperature and salt presence conditions. The issue is addressed rigorously, with well- demonstrated knowledge of the literature on this topic and the use of appropriate modern methods. They produced new knowledge in the field and reached the conclusions, which are well supported by the presented results. Under these circumstances, the reviewer recommends the acceptance of the paper, pending some relatively minor comments, which would be best addressed in the form of discussion. Addressing these comments should improve the paper with respect to clarity and the impact on readers.

Minor comments

1) Overall the writing style is somewhat “choppy”, but clear and there is no need to make any changes in this regard.

Answer: Thanks for the comment. We revised the manuscript according to your following comments.

2) P. 2, 3rd paragraph, line 4 – accumulation of chloride in settling tanks (as in stormwater ponds) would cause stronger densimetric effects than similar.

Answer: You are right, it is now clearly described.

3) Fig. 1b – one influential factor missing is particle flocculation (studied e.g. by Krishnappan, B.G. and Marsalek, J. (2002). Modelling of flocculation and transport of cohesive sediment from an on-stream stormwater detention pond. Wat. Res., 36(15): 3849-3859; or Krishnappan, B.G., Marsalek, J., Watt, W.E., Anderson, B.C. (1999). Seasonal size distribution of suspended solids in a stormwater management pond. Wat. Sci. tech., 39(2), 127-134). In other words, the authors should state up front that they dealing with “discrete” particle settling. Flocculation would be typical for warmer seasons and contributes to greater sizes of particles in eq.(6).

Answer: Thanks for that helpful comment. We discussed that topic as follows in the manuscript and added new references:

“Furthermore, studies showed that the flocculation of PM is affected by the temperature [21-23]. However due to a decreasing floc density with an increasing floc size, it is not clear if flocculation is definitely improving the settling behavior. Thus, this study is describing discrete particle settling.“

4) P. 3, 2.1, 2nd paragraph – Flow measurement – what was the accuracy of this instrument? Note that the threshold inflow was extremely low - just 1.67 x 10-5 (L/s).

Answer: The threshold value was 1 L/min = 0.017 L/s. For Q > 6.3 L/min = 2.6 L/s/ha the error is < 1.6 %. That info was added in the manuscript. Below the 2.6 L/s/ha the error will increase, but there are no values available for that. Only 3 of the 23 rain events had an average flow below this value.

5) “To prevent alteration of samples, they were kept in coolers at 4+/-1 C…

Answer: Tanks for the correction. It was adopted.

6) The same section, paragraph 4: “ were combined to create one composite sample…” – this is a better word.

Answer: Thanks for the hint. I changed the word.

7) The same paragraph – was the upper limit of particle sizes in TSS samples applied?

Answer: This sentence was added: Large constituents (>> 1 mm; e.g. leaves, cigarette stubs), which are not representative for the sample, were manually removed.

8) P. 5, eq. (8) – this is also known as the equation describing the “ideal settling basin” introduced by Camp.

Answer: Unfortunately, we were not able to access that fundamental study.

9) P. 6, the paragraph below Table 1, please clarify the reasoning why “Furthermore, in summer, some events showed a low TSS retention.”

Answer: We described the section now more precise. However, reasons for the low retention during the other two events are not derivable from the data.

10) Fig. 2, could high retentions of TSS be influenced by particle flocculation?

Answer: That could be one reason. I added Fig. 3 which is showing a approximation of the PSD. In this graph the increasing particle size is explained by increased flocculation.

11) The first paragraph under Fig. 2, please reword “However, the SS63 parameter has become popular only…”

Answer: Thanks for the comment. We reworded it.

12) Paragraph 1 below table 2, the reader would benefit from seeing some explanation of “the influent hydrograph unsteadiness”.

Answer: We added a brief description of the parameter.

13) Notation of Figs. 4-8, units indicated by fractions (e.g., d /µm) – somewhat unusual. Are they acceptable in this journal? I am sued to seeing units in the brackets, e.g., d(µm).

Answer: The units are now indicated in brackets, e.g. d (µm).

14) TSS retention efficiency – this parameter would deserve a bit of discussion of uncertainties in the present values. Such concerns were expressed e.g. by Bardin, J.P., Barraud, S. Chocat, B. (2001). Uncertainty in measuring the event pollutant removal performance of online detention tanks with permanent outflow. Urban Water, 3(1-2), 91-106.

Answer: Thanks for that crucial advice and very good reference. Uncertainty cannot be assessed unproblematically due to the lack of possibility to reproduce rain events and different sampling systems. We added therefore a “warning” with a reference to appropriate literature. Furthermore, we conducted a short approximation of uncertainty induced by TSS analysis error to underline the issue.

15) P. 17, 2nd line, “sedimentation tank” was earlier called a shaft.

Answer: The system is now called “sedimentation tank” within the whole manuscript.

16) German references – it is my experience that the authors need to identify the language, and an English translation should follow the title in the original language.

Answer: According to the editor, we added translated titles in the reference list.

Reviewer 2 Report

This paper describes the impact of temperature and de-icing salt on the sedimentation of particulate matter in traffic area runoff. The sampling of SQID's can be difficult at the best of times however during cold/frosty events there are definitely different processes at work. Most have been adequately explained in this paper even if the data was insufficient, however I believe there are other processes that were not discussed. For example, the impact of PSD of the frost/ice on entry to the SQID. I have observed ice/snow intermittently block inlet points to SQID's during rain events, which causes a by-pass effect (decrease in flow and TSS load to SQID during that specific event). Could this potentially occur at your site? If it does then what impact does this have on your interpretation? (build larger surface area basins). The authors state, referring to Spelman & Sansalone (2018) "they identified influent hydrograph unsteadiness as the most powerful influencing factor." Continuous simulation of many events coupled with a mass-balance approach may have provided a more realistic view of actual processes at this site. Also, the authors mention scouring of the sedimentation basin at the start of the sampling campaign.....this indicates to me that the overall performance of the SQID is relatively low due to existing maintenance cycles (timely removal of sediment from the tank). Maintenance cycles should ideally be rainfall event based, i.e, make sure they are cleaned out prior to larger rain events that typically cause scouring. While the paper uses fundamental principles to describe the sedimentation processes (which are fine), the lack of hydrological data and stationary method approach means this paper does not reach the heights it should. If modeling/discussion could include PSD of ice/snow during a site-specific event and also provide continuous simulation of several events then this paper would be very much improved.

Author Response

Reviewer 2

Comments:

1) This paper describes the impact of temperature and de-icing salt on the sedimentation of particulate matter in traffic area runoff. The sampling of SQID's can be difficult at the best of times however during cold/frosty events there are definitely different processes at work. Most have been adequately explained in this paper even if the data was insufficient, however I believe there are other processes that were not discussed. For example, the impact of PSD of the frost/ice on entry to the SQID.

Answer: Thanks for the comment. Unfortounately, it was not possible to measure the PSD during that monitoring. We will consider that in future studies. To approximate the PSD, we used the ration SS63/TSS. And plotted and discussed it in Fig. 3 on p.7.

2) I have observed ice/snow intermittently block inlet points to SQID's during rain events, which causes a by-pass effect (decrease in flow and TSS load to SQID during that specific event). Could this potentially occur at your site? If it does then what impact does this have on your interpretation? (build larger surface area basins).

Answer: We never observed snow or frost inside the system. However, catchment area and hydrograph characteristics may change due to different flow paths on the road surface. But considerable amounts of snow on the road surface is a rare event in Munich and during those no sampling was possible because of the low discharge rates.

3) The authors state, referring to Spelman & Sansalone (2018) "they identified influent hydrograph unsteadiness as the most powerful influencing factor." Continuous simulation of many events coupled with a mass-balance approach may have provided a more realistic view of actual processes at this site.

Answer: Thanks reviewer for that legitimate objection. Scope of this study was to assess fundamental influences of temperature and de-icing salt on the sedimentation of PM. As noted in the manuscript (p. 12, second to last paragraph, last sentence) further studies should use numerical modeling and non-steady state conditions to determine the effects more closely. However, to achieve a reliable modeling and to be able to validate the results, temporal variations of TSS in the system need to be measured. The data used in the present study is just based on event mean concentrations, therefore at this stage a numerical modeling is not feasible.

4) Also, the authors mention scouring of the sedimentation basin at the start of the sampling campaign.....this indicates to me that the overall performance of the SQID is relatively low due to existing maintenance cycles (timely removal of sediment from the tank). Maintenance cycles should ideally be rainfall event based, i.e, make sure they are cleaned out prior to larger rain events that typically cause scouring.

Answer: Out of a scientific perspective that is true to investigate detailed processes of the sedimentation. However, shorter maintenance cycles would increase operational costs to a level risking economic feasibility of SQIDs. Aim of the monitoring was to reflect realistic conditions and therefore the tank was not cleaned after each event.

5) While the paper uses fundamental principles to describe the sedimentation processes (which are fine), the lack of hydrological data and stationary method approach means this paper does not reach the heights it should. If modeling/discussion could include PSD of ice/snow during a site-specific event and also provide continuous simulation of several events then this paper would be very much improved.

Answer: See our comment on modeling above (3). The PSD was not measured during this monitoring. However, it can be approximated by the ratio SS63/TSS. To illustrate variation in PSD I added Fig. 3 on p. 7.